# Identifying Optimal Cell Size for Geodiversity Quantitative Assessment with Richness, Diversity and Evenness Indices

Catarina Lopes [1,*], Zara Teixeira [2], Diamantino I. Pereira [1] and Paulo Pereira [1]

1 Institute of Earth Sciences, University of Minho, Gualtar Campus, 4710-057 Braga, Portugal
2 MARE-Marine and Environmental Sciences Centre, ARNET-Aquatic Research Network, Department of Life Sciences, University of Coimbra, 3000-456 Coimbra, Portugal
* Correspondence: catas.lopes@gmail.com

**Abstract:** The importance of quantitatively assessing the spatial patterns of geodiversity, and their intrinsic relationship with biodiversity and the ecosystem services provided to society, has been signalized by several authors, due to the relevance of this information in territorial management, the planning of environmental and conservation strategies. Within geodiversity method assessments, the grid system is the most widely used GIS spatial approach to calculate a geodiversity index. Preferred for its simplicity, it implies the fundamental decision of choosing the scale of the analysis, defined by the selection of cell size, determinant for the accuracy and correctness of the final maps. Although this topic has been occasionally approached by some authors within geodiversity assessments, there is no formal procedure for cell size selection. This is a key issue, and, in the scope of the present work, an empirical procedure to select optimal cell size(s) was tested on the national scale in Portugal, in lithology and geomorphology datasets. The quantitative method based on geodiversity indices was applied, using richness, diversity and evenness indices, in a hexagonal analytical grid, through eight cell dimensions. Several descriptive statistical parameters were analyzed, with particular emphasis on dispersion statistical measures. Optimal cell size corresponded to the minimum cell size, once dispersion values were significantly reduced or stabilized, and distributions from evenness and diversity indices were closer to symmetry, which provided more accurate results and higher spatial differentiation, although the final decision should always consider the main purposes of the analysis.

**Keywords:** grid system; geodiversity quantitative assessment; cell size; diversity indices

## 1. Introduction

Geodiversity is often defined as the abiotic equivalent of biodiversity, described as "the natural range (diversity) of geological (rocks, minerals, fossils), geomorphological (landform, topography, physical processes), soil and hydrological features. It includes their assemblages, structures, systems and contributions to landscapes" [1].

However, many other definitions of geodiversity exist that differ on the type and number of abiotic elements considered, complexity and scope of the concept, methodological procedure and even on the type of data [2–6].

The need to quantitatively assess the spatial patterns of geodiversity, as well as its intrinsic relationship with biodiversity and the ecosystem services provided to society, has been emphasized by several authors, given the imperative value of this information within territorial management and the planning of environmental and conservation strategies. In light of the new COP15 "30 × 30" agreement ((https://www.cbd.int/article/cop15-final-text-kunming-montreal-gbf-221222 (accessed on 12 May 2023)), which calls for protecting 30 percent of the world's terrestrial and marine habitats by 2030, this knowledge has become even more critical and imperative, since geodiversity, as a surrogate for biodiversity, might have a determinant role in the definition of those areas.

Geodiversity assessment methodologies can be divided according to: (i) the type of data (direct and indirect methods) [7]; and (ii) the methodological procedure (qualitative,

qualitative–quantitative and quantitative) [8]. Direct methods are based on pre-elaborated thematic maps that constitute the input data for GIS treatment, commonly the computation of richness of geodiversity elements. Indirect methods, also called 'surrogate indicators', usually apply geomorphometric methods, based on DEM and derived products, to calculate the physical properties of the terrain. The qualitative approach is based on a description of the geodiversity elements of a given area, usually applied on a local or site scale; their specificities; and values, defined by experts, within which a link can be established with geoheritage evaluation and ecosystem services qualitative assessment [9]. Within the hybrid qualitative–quantitative methods, advanced technical solutions for assessing geodiversity have been developed, and are generally built on map algebra, analytic hierarchy process and multicriteria spatial analysis procedures, fundamentally requiring the consultation of expert knowledge [10–15]. Quantitative methods are the most common approach and encompass different procedures, namely indices, landscape metrics, statistical modelling, map algebra, etc., among which geodiversity index mapping, and in particular the grid system, is the most used [16]. It results from the sum of partial indices of lithology, geomorphology, hydrology and pedology, and may also embrace other abiotic elements, depending on the main goal of the research. The most cited and applied geodiversity index [4] relates the variety of physical elements (geology, geomorphology, hydrology and soils) with surface roughness. Pereira et al. [17] popularized a methodology based on the grid system, widely applied, and developed under the coordination and collaboration of the Portuguese authors, mostly in Brazil [18–22]. Apart from the original geodiversity index based on kernel density developed by Forte [23] and tested on a local scale (in the Mafra municipality), and the richness geodiversity index presented on a national scale by Peixoto [24], there are no significant geodiversity studies applied to the Portuguese territory.

The geodiversity index mapping generally relies on richness, i.e., the number of distinct classes, within a certain area, measured in a predefined unit (unit cell within the grid system), which in turn reflects the "primary geodiversity" or "intrinsic geodiversity" of that area [25]. (The term "intrinsic" used by Carcavilla [25] has a distinct meaning from the "intrinsic value" of geodiversity proposed by Gray [26], which is related with the ethical belief that some things (in this case the geodiversity of nature) are of value simply for what they are rather than what they can be used for by humans (utilitarian value)). In order to approach the wide complexity of geodiversity, some authors have highlighted the necessity of complementing a richness assessment with other parameters (e.g., frequency, spatial distribution) [25], by additionally pointing out the fact that richness corresponds solely to one of two primary components that form the concept of diversity, with the other being the relative amount of each distinct class [27]. From this perspective, there is a clear lack of studies providing a broad evaluation of geodiversity spatial patterns, despite the existence of a few works that have adopted composition diversity (and other) metrics [28–34]. Composition diversity metrics are influenced by two components—richness and evenness, generally referred to as the compositional and structural components of diversity, respectively [35]. Richness indicates the number of different classes, indifferent to the relative abundance of each class type or the spatial arrangement of classes. Evenness measures the relative abundance of different class types, emphasizing either relative dominance or evenness. Diversity is, thus, a composite measure of richness and evenness. Richness, Shannon's Diversity Index (SHDI), Shannon's Evenness Index (SHEI) [36], Simpson's Diversity Index (SIDI) and Simpson's Evenness Index (SIEI) [37] are the most common indices used to measure composition diversity.

The use of the grid system, chosen by many for its simplicity, clarity and adaptability, implies a fundamental and determinant decision—the selection of the cell size of the analysis grid, i.e., the scale of the analysis—which, in turn, should comply with the inherent properties of the input datasets, being a function of the main goals of the analysis [38]. Unlike the absolute representation of space, which is related to the standard geographic reference systems, the scale of the analysis corresponds to a relative representation of space, which is driven by more arbitrary and less rigid rules that are more difficult to define [39].

Many authors have studied this problematic, either by discussing and presenting analytical procedures, or by testing the effect of distinct cell sizes on their (modelling) analysis [30,38,40–44]. Among these authors, Hengl [38] is among the most cited within geodiversity studies. He presents some empirical and analytical rules for the selection of the basic unit size (grid resolution), based on cartographic (study area, working scale, positional accuracy, inspection density, size of objects, distance between points, spatial dependence structure and complexity of terrain); statistics (best predictive properties, i.e., the pixel size that offers best correlation coefficient with the main variable); and information theory concepts, proposing for each type of rule three standard grid resolutions (coarsest, finest and recommended), indicating the corresponding formula. Some of these rules are not so easy to apply, requiring some (advanced) processing. Within geodiversity studies, many authors adopt the simplest rule proposed by Hengl [38] to define the cell size (either for the analysis grid and/or grid resolution), by applying the scale factor based on a given set of input layers. Many others use arbitrary cell sizes from published works, usually based on scale factor or surface area. Other authors assume a more empirical methodology, testing diversity measurements (usually richness, but also SHDI) within several cell dimensions, comparing and analyzing the results e.g., [19,30,45]. These approaches based on diversity measurements consider simultaneously the properties of the input dataset and the goal of the analysis, resulting in the simplification of the input cartographic dataset into a diversity analytical grid. Within this context, Pereira et al. [17] presented an empirical solution, frequently adopted by other authors, that defines the optimal cell size of the analysis grid as the one providing simultaneously the maximum range of values and the lower "minimum" value, i.e., a smaller cell size would provide lower maximum values, while a larger cell size would lead to higher minimum value. Along these lines, Eiden et al. [41] established the optimal cell size based on the maximum range of values that indicate simultaneously "the optimum spatial differentiation of the territory and the maximum range of diversity measures". For larger cell sizes, the resulting map would show for almost cells a maximum of diversity and would not provide a relevant spatial differentiation. At the other extreme, a minimum cell size would produce a resulting map with very low values all over the area, and the higher values would be restricted to the borders between classes [41]. Eiden et al. [41], who applied these tests on the European level, highlighted the importance of a more in-depth investigation, with more empirical tests of the optimal cell size. Bartuś [30] proposed a procedure to determine the optimal cell size by analyzing distribution and some descriptive statistical parameters, in the analytical grid, of richness and SHDI of lithostratigraphic map of the Ojców National Park (Poland), on a medium-scale landscape.

Clearly being a key issue within geodiversity assessment methodologies, the present work presents an empirical methodology to select the most appropriate cell size to assess geodiversity by using the lithology and geomorphology data of mainland Portugal as a case study. These results are part of an ongoing study of the geodiversity assessment of mainland Portugal, and are key for analyses comparing different geodiversity elements and relations with other natural and cultural features. Broader scales (national, regional), although not commonly applied in geodiversity assessments, have been used by some authors [10,17,24,28,46], providing an overall quantification of the physical heterogeneities of geodiversity by highlighting the major features of the territory, which, in turn, can provide relevant information for territorial management and the planning of environmental and conservation strategies, as well as for establishing a correlation with other major physical factors and territory features, land cover and land use, and with major spatial biodiversity patterns, since some studies indicate that geodiversity may be an important correlation of biodiversity at landscape, subnational scales [47–49].

Following the work of Bartuś, T. [30] and Eiden, G. et al. [41], the direct quantitative method based on geodiversity indices is applied, by using richness, Simpsons's and Shannon's diversity and equity indices (SIDI, SIEI, SHDI, SHEI), to the lithology (1:1,000,000) and geomorphology of mainland Portugal (1:500,000), in a hexagonal analytical grid, through eight cell dimensions (1 km, 2 km, 5 km, 10 km, 15 km, 20 km, 25 km and 30 km). The

hexagonal cell grid provides more natural pattern results, as it attenuates the edge effect of the grid shape, due to its circularity, i.e., low perimeter to area ratio, constituting the most circular-shaped polygon that might form a continuous evenly spaced grid. Several descriptive statistical parameters are analyzed along the various dimensions of the unit cell, as a method of evaluation/selection of the most adequate dimension for the analysis. Additionally, the correlation between the indices along the different cell dimensions is analyzed with the Spearman and Pearson correlation factors. Finally, the effect of cell size on the final maps of lithological and geomorphological diversity is also analyzed, using the conventional representation of the five classes (very high, high, medium, low and very low), based on the Jenks classification, by evaluating the area occupied per each class along the distinct cell sizes.

## 2. Cell Size Analyses

### 2.1. Study Area

2.1.1. Geology

Several authors studied in detail the structural evolution of the geological formations of Portugal [50–57]. The geology of Portugal is considered very diverse and complex, resulting from the involvement of the Portuguese terrains in several geological cycles (Cadomian, Hercynian and Alpine/Atlantic), which produced successive geodynamic environments and originated diverse and important geological resources [58]. For the present work, the Geological Map of Portugal at a 1:1,000,000 scale by the National Laboratory of Energy and Geology [59] was used (Figure 1). This map, also referred as a lithochronological map, corresponds to a global and synthetic view of the lithological diversity and structural complexity of mainland Portugal's geology [58].

Regarding the main geotectonic units and structures, Portugal is divided into: (i) the Iberian Massif, or pre-Mesozoic substrate that corresponds approximately to 70% of the mainland territory; and (ii) the Meso-Cenozoic basins, which include the western Lusitanian Basin, the southern Algarve Basin and the Tagus and Sado basins. The pre-Mesozoic substrate, dominated by granitoids and Cambrian and Precambrian metasediments, is classified into four large lithostructural zones, each of them with its own geological history: the Galicia-Trás-os-Montes Zone (GTZ); the Central-Iberian Zone (CIZ); the Ossa-Morena Zone (OMZ); and the South-Portuguese Zone (SPZ) [58]. GTZ is described by the existence of two mafic and ultramafic polymetamorphic massifs, known as the Bragança and Morais Massifs, with surrounding formations characterized by the existence of acid and basic volcanic rocks, which contact the massifs via larger thrust systems [60,61], corresponding to allochthonous complexes (with allochthony > 50 km) (dating from Neoproterozoic to Devonian) and associated para-autochthonous terrains (Ordovician-Devonian) [58]. The CIZ, OMZ and SPZ present autochthonous, sub-autochthonous and allochthonous terrains (with allochthony < 50 km). The CIZ is mainly characterized by the predominance of the schist–greywacke complex formations (Dúrico-Beirão Supergroup), consisting of phyllites, metagreywackes, metaquartzowackes, metaconglomerates, metalimestones and schists (flysch), as well as gneisses and migmatites, a flysch-type series dating from the Cambrian and Ediacaran period [58,60,61]. There are also large areas of granitoids, in which various types of granite can be distinguished, representing episodes of magmatism related to the Hercynian cycle. Quartzites from the Ordovician also occur within CIZ. The OMZ is an extremely complex and diverse unit, which begins with a polymetamorphic Precambrian formation belonging to the Cadomian sock, followed by Cambrian, Ordovician and Silurian formations, and ends with a flysh sequence from the Devonian period, presenting oceanic crust in the Upper Devonian and in the Ordovician periods [58,60,61]. This unit contacts with the CIZ via an important shear zone passing by Tomar and Badajoz (Blastomylonitic Belt), which stretches from Oporto to Cordoba in Spain [60,61]. Magmatism related to the Hercynian cycle is also present in the OMZ. In both the OMZ and the CIZ, these are represented by granitic rocks, gabbros, diorites, tonalites, anorthosites, granodiorites, diorites, migmatitic gneisses, granitic orthogneisses, peralkaline metasienites and peridotites.

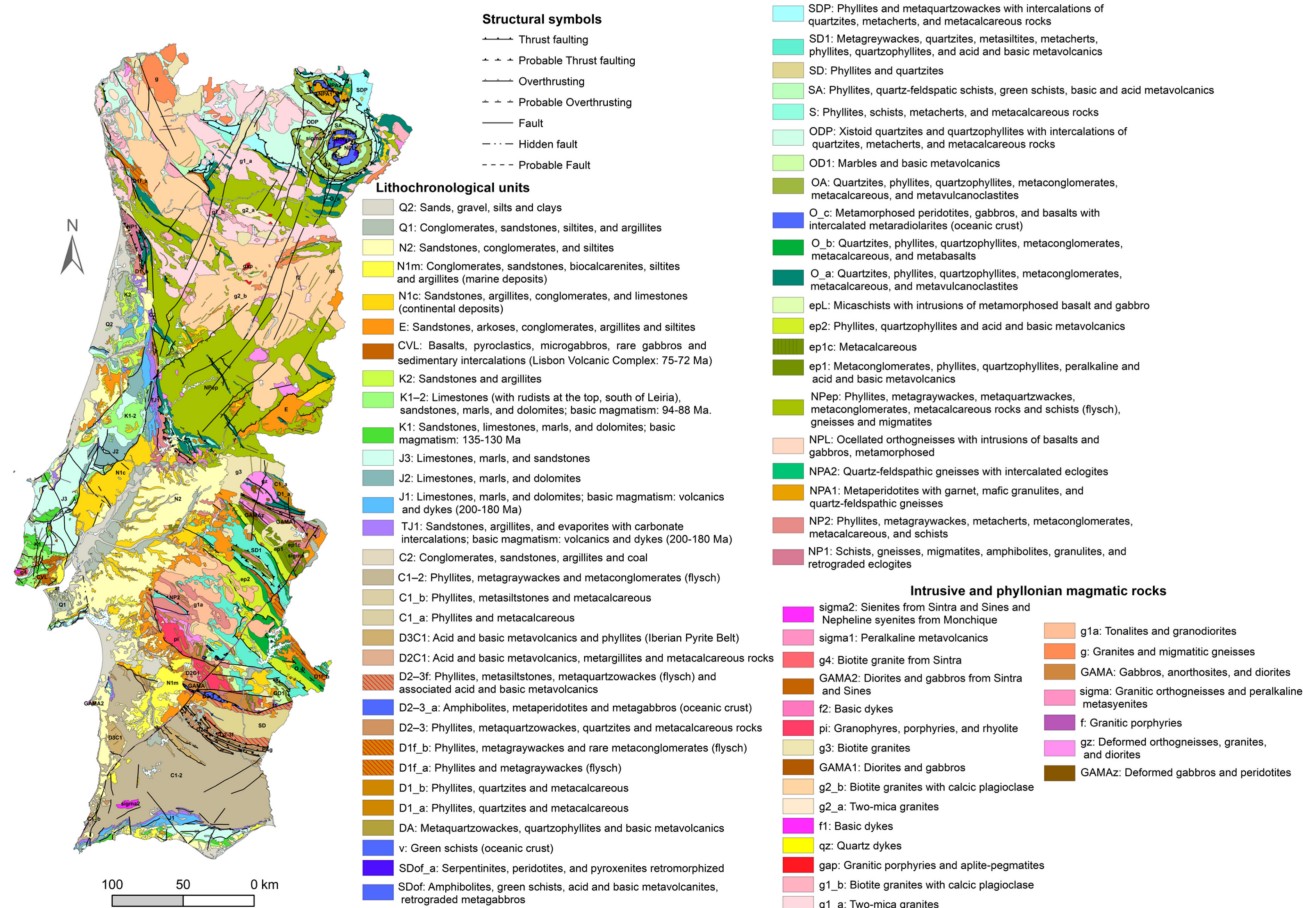

**Figure 1.** Geological map of mainland Portugal at a 1:1,000,000 scale produced by the National Laboratory of Energy and Geology [58,59,62], corresponding to a synthesis of the current geological knowledge of the territory, here representing the emersed and continental region. The original can be found at https://geoportal.lneg.pt/media/p4wft3w5/cgp1m_2010.pdf (accessed on 12 May 2023). The map legend presented here shows the lithochronological units "ordered" from the most recent to the oldest. The complete lithostratigraphic map legend adapted and translated to English from the original, can be found in the Supplementary Materials (Figure S1).

The Ferreira-Ficalho thrust (E-W to the east and NW-SE to the west) marks the border between the OMZ and SPZ. The SPZ is characterized by the existence of a volcanic sedimentary complex (the Iberian Pyrite Belt) from the Late Devonian–Earlier Carboniferous period, overlaid by a flysch sequence; underlying this complex is the so-called "Phyllite-Quartzite Group". The Iberian Pyrite Belt is the most important metallogenic province in Portugal. The "Pulo de Lobo" is the oldest formation of this zone, which includes phyllites, quartzites and rare acid and volcanic rocks [60,61].

The geodynamic evolution of Iberia in the Meso-Cenozoic is dominated by the Tethys/Atlantic cycle and corresponds to the formation of the Western (Lusitanian) and Southern (Algarve) basins (borders), as well as the Lower Tagus and Sado basins [60,61]. The Meso-Cenozoic sedimentary basins (with the oldest formations dating from the Upper Triassic) were mainly filled by evaporites, limestones (mainly from the Jurassic, but also from the Cretaceous), dolomites, clays, marls, sandstones and volcanites [58]. Significative magmatism related to the Alpine/Atlantic cycle also occurred between 100–72 Ma, comprising syenites from Sintra and Sines and nephelinic syenites from Monchique, biotitic granite from Sintra and diorites and gabbros from Sintra and Sines [58].

The lithochronological map presents two main attributes (lithochronologic unit and age) displayed by geotectonic zone, from which the lithochronologic unit was used to produce the map of lithological diversity. This map is composed of 74 (unique) multifeatures,

which correspond to 2209 single features. There is a superior diversity of ages and unique features in the OMZ, immediately followed by the Meso-Cenozoic basins. The number of unique features in GTZ is notable, considering the relative amount of area (7%) (Table S1 and Figure 1).

### 2.1.2. Geomorphology

Apart from publications concerning the geology of Portugal, specific works on the geomorphology of the Portugal mainland were analyzed, namely the work of Ribeiro et al. [63]; the geomorphological map of Portugal [64], with a linear representation of landforms (related to coastal, river, and glacial processes); works on the major regional relief units [65]; a major physical geography synthesis [66]; a hierarchical classification of geomorphological units [67]; and a review on the landscape of Portugal [68]. The work of Pereira et al. [67] and its improvement into a more detailed map of geomorphological units (Figure 2) were used in this work to support the methodological procedures.

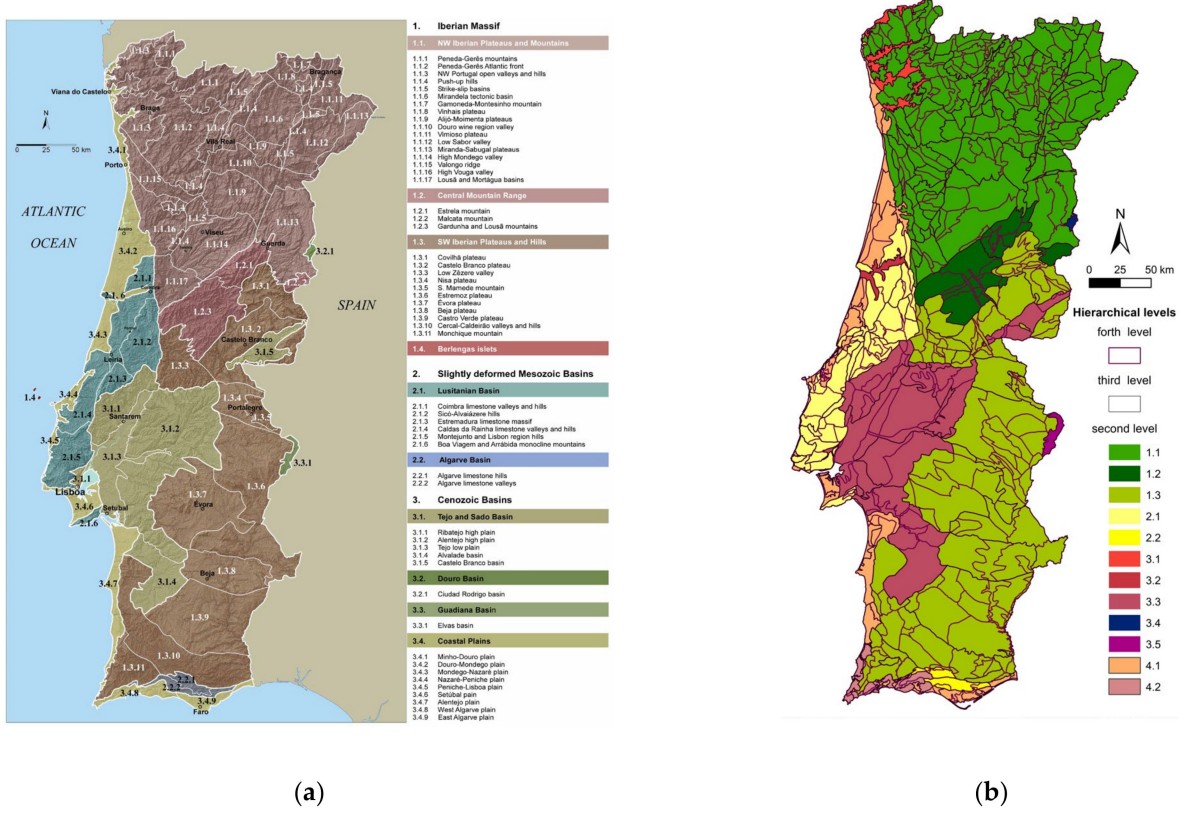

(**a**)           (**b**)

**Figure 2.** Geomorphological units of mainland Portugal: three-level geomorphological units [67] (**a**); map developed in the present work, by detailing and reorganizing the previous version, displaying four-level geomorphological units, at a 1:500,000 scale (**b**).

According to the classification proposed by Pereira et al. [67], Portugal mainland territory is compartmentalized into three hierarchical levels of consistent and homogeneous geomorphological information (attending to dimension, age and genesis), namely morphostructural (3 units), morphoscultural (10 units) and morphoscultural sub-units (56 units). The first level corresponds to the large Iberian Peninsula morphostructural units: the Iberian Massif; Slightly Deformed Meso-Cenozoic Basins (Lusitanian and Algarve); the non-deformed Tejo and Sado Cenozoic basins. Within the Iberian Massif, the major tectonic structures are predominantly oriented in a NW-SE direction, aligned with the Ordovician quartzite ridges, but also fini-Hercynian faulting is strongly printed in the relief, with N-S, NNE-SSW, NE-SW and NW-SE orientations. The hydrographic network organization is passively controlled by this fracture network. The slightly deformed Mesozoic basins

correspond to the Lusitanian and Algarve basins and cover 8% of the national territory (7% Lusitanian Basin; 1% Algarve Basin), being frequently covered by unconsolidated or poorly consolidated Cenozoic sediments, namely in the "Coastal Plains". The Cenozoic basins are mainly represented by the Tejo and Alvalade Cenozoic basins, corresponding to 15% of the national territory. The Cenozoic basins of Douro and Guadiana have a small extent in Portugal. These large morphostructural units are divided into morphoscultural units, which correspond to compartments initially generated by tectonic processes and modeled by weathering processes over geological time [67]: the Iberian Massif (NW Iberian Plateaus and Mountains, Central Mountain Range, the SW Iberian Plateaus and Hills and Berlengas); Slightly Deformed Meso-Cenozoic Basins (Lusitanian Basin, Algarve Basin); Cenozoic Basins (Baixo Tejo and Alvalade, Douro, Guadiana, Coastal Plains). The second level morphoscultural units are divided into 56 morphoscultural sub-units, which correspond to compartments modeled by specific weathering processes, delimited based on the analysis of relief patterns provided by the SRTM datasets, geological substrata and fieldwork. The map developed for this work (Figure 2b) contains a fourth level of more detailed geomorphological units and a reorganization in the limits of the upper levels. This fourth level corresponds to individualized landforms related to aggradation, erosion or denudation, such as river or marine plains, river or marine terraces or hills and ridges, for a total of 686 features (Table S2 and Figure 2b) representing the geomorphological diversity of Portugal. The morphoscultural unit Mountains and Plateaus of the NW Iberia comprises the highest number of features, followed by the Plateaus of the Peninsular SW and Coastal Plains units. The Peneda-Gerês mountains, the Peneda-Gerês Atlantic Front and Mirandela tectonic basin morphoscultural sub-units contain the highest number of individualized fourth-level landforms. In the SW Iberian Plateaus and Hills morphoscultural unit, it is in the Castelo Branco Plateau morphoscultural sub-unit where the highest number of fourth-level units can be identified. The Coastal Plains sub-unit, although not so extensive in area, has a very significant number of distinct features.

### 2.2. Methodology

Aiming to identify the optimal cell size(s) to assess the geodiversity of Portugal mainland (national scale) by using the grid system, richness (RICH), Simpsons and Shannon diversity and equity indices (SIDI, SIEI, SHDI, SHEI) of lithology (1:1,000,000) and geomorphological units of mainland Portugal (1:500,000), were calculated in a hexagonal analytical grid, along eight cell dimensions (Tables 1 and 2). Hexagonal cell grids provide more natural pattern results, as they attenuate the edge effect of traditional grid shapes, due to their circularity, i.e., a low perimeter to area ratio, constituting the most circular-shaped polygon that might form a continuous evenly spaced grid. Several descriptive statistical parameters were analyzed along the various dimensions of the unit cell, as a method of evaluation/selection of the most adequate dimension for the analysis. Additionally, the effect on the correlation between the indices along the different dimensions of cells was analyzed, through the Spearman and Pearson correlation factors. Finally, the effect of cell size on the final maps of lithological and geomorphological diversity is also analyzed, using the conventional representation of the five classes (very high, high, medium, low and very low), based on the Jenks classification, by evaluating the area occupied per each class along the distinct cell sizes.

Considering the formula of the indices and correspondent domain of values (Table 2), some considerations must be taken into account for the interpretation of results. Richness measures the number of categories, expressed in integers, ignoring their relative abundance or spatial arrangement, i.e., the same richness value can correspond to a range of distinct evenness values, except in a case where a single category is present. Evenness measures the distribution of area among patch (category) types where larger values imply greater diversity, independently of richness. Evenness equals zero when a single category is present (distribution of area among different categories extremely uneven, dominated by one single category). Evenness equals 1 when the observed diversity equals perfect evenness, i.e., a

proportional abundance of each category. The quantification of evenness derives from a corresponding diversity index. Diversity indices increase with the number of categories (richness) and/or an increase of the proportional distribution of area among different categories, although being more sensitive to richness than to evenness. Diversity equals 0 when only one category is present (no diversity). SHDI, the most popular diversity index, is based on information theory, representing the amount of "information" per class [35]. The value of SIDI represents the probability that any two patches selected at random will be different types: the higher the value, the greater the diversity [35]. The four diversity and evenness indices are expressed in decimal values.

**Table 1.** Hexagonal cell grid main characteristics used to assess lithological and geomorphological diversity of mainland Portugal: cell dimensions, area per cell unit, total number of cells per dimension (*n*), number of empty and filled cells per dimension.

| Cell_km (L) | Area Cell km$^2$ | *n* Total | Empty | Filled |
| --- | --- | --- | --- | --- |
| 30 | 779.4 | 345 | 195 | 150 |
| 25 | 541.3 | 442 | 230 | 212 |
| 20 | 346.4 | 620 | 305 | 315 |
| 15 | 194.9 | 1066 | 530 | 536 |
| 10 | 86.6 | 2379 | 1223 | 1156 |
| 5 | 21.7 | 9516 | 5150 | 4366 |
| 2 | 3.5 | 59,170 | 32,780 | 26,390 |
| 1 | 0.9 | 236,070 | 131,848 | 104,222 |

**Table 2.** Richness, diversity and evenness indices' formulas used to assess lithological and geomorphological diversity of mainland Portugal: *m* corresponds to the number of unique categories per unit area and *Pi* correspond to the proportion of area occupied by a category of type *i*.

| Index | Formula | Range |
| --- | --- | --- |
| Richness (Rich) | $\text{Rich} = m$ | $\text{Rich} > 0$ |
| Shannon's Diversity Index (SHDI) | $\text{SHDI} = -\sum_{i=1}^{m}(P_i \ln P_i)$ | $\text{SHDI} \geq 0$ |
| Shannon's Evenness Index (SHEI) | $\text{SHEI} = \frac{-\sum_{i=1}^{m}(P_i \ln P_i)}{\ln m}$ | $0 \leq \text{SHEI} \leq 1$ |
| Simpson's Diversity Index (SIDI) | $\text{SIDI} = 1 - \sum_{i=1}^{m} P_i^2$ | $0 \leq \text{SIDI} \leq 1$ |
| Simpson's Evenness Index (SIEI) | $\text{SIEI} = \frac{1 - \sum_{i=1}^{m} P_i^2}{1 - \left(\frac{1}{m}\right)}$ | $0 \leq \text{SIEI} \leq 1$ |

Both maps have distinct scales and significative heterogeneous polygon areas with non-uniform distribution along Portugal mainland. Lithology is represented by a significant diversity of polygon areas and forms, displaying bigger and smaller features than those from geomorphology. The polygon area varies from 0.00016 km$^2$ to 6838.4 km$^2$ (coefficient of variation: 6.4). Geomorphology units exhibit, at the fourth hierarchical level of information, polygon areas varying from 0.12 km$^2$ to 1337.32 km$^2$ (coefficient of variation:1.42).

Several descriptive statistical parameters were analyzed along the eight cell dimensions for each map, as potential indicators for the optimal cell size, with particular emphasis for dispersion statistical measures, namely the quartile coefficient of dispersion, coefficient of variation, and skewness coefficient, range, min, max and interquartile range (IQR). Additionally, the correlation factors of Spearman and Pearson were calculated to assess the effect of cell size on the correlation between the indices. Lastly, final maps of lithological and geomorphological diversity based on the five indices were produced for each cell dimension, using the conventional representation with five classes (very high, high, medium, low and very low), based on the Jenks natural breaks classification, which provided the

best differentiation between classes. The area occupied per each class along the distinct cell sizes was then analyzed. GIS processing, uni and bivariate statistic calculations were performed in QGIS 3.16.16 and Andad 7.12 (CVRM/IST).

## 3. Results

### 3.1. Lithological Indices–Statistical Parameters

#### 3.1.1. Mode, Range, Min, Max and IQR

Richness tends to vary widely with cell size (Figure 3). This is especially observable in the correspondent maximum value, i.e., the maximum number of distinct categories, which shows a faster progression in the first cell sizes, until 10 km, then stabilizing at around 11–12 categories, between cell sizes of 10 km and 20 km, and increasing significantly to its maximum, 17 categories, present in both of the biggest cell sizes, 25 km and 30 km, indicating another platform of values.

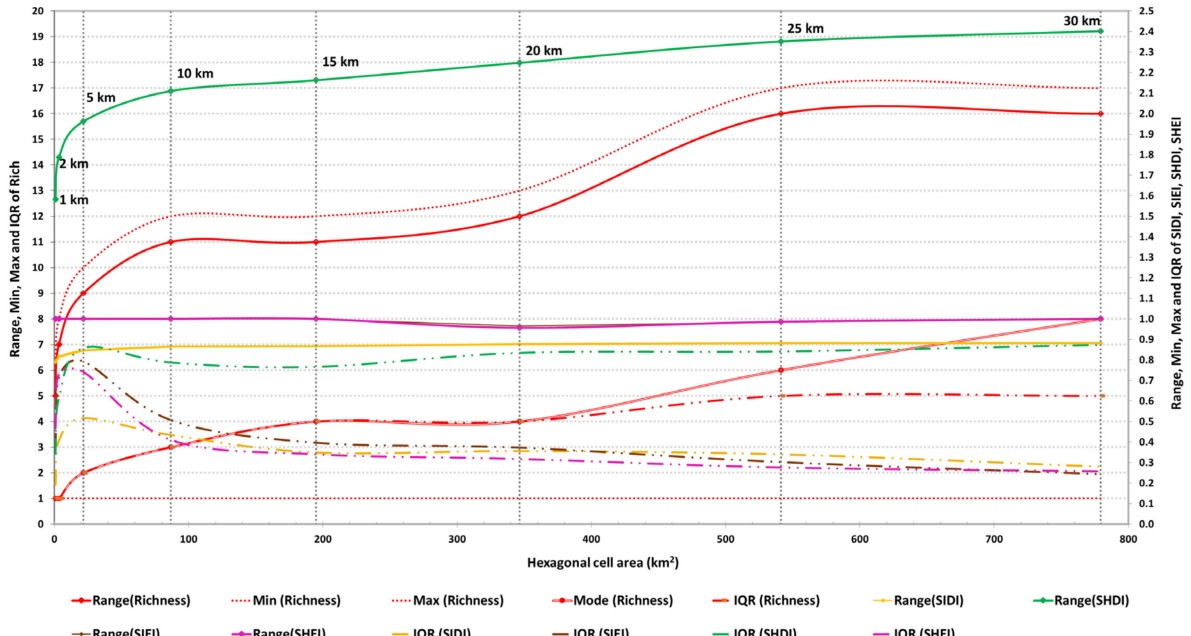

**Figure 3.** Mode (richness), range, min, max and interquartile amplitude (IQR) of richness, SIDI, SIEI, SHDI and SHEI, calculated in a hexagonal analytical grid, along eight cell dimensions (1 km–30 km), intersected with lithology data from the geological map of mainland Portugal at a 1:1,000,000 scale. SIDI, SIEI, SHDI and SHEI present much smaller values projected in the secondary axe of the graphic. Since the minimum value (min) is constant through all cell sizes for all indices (1 for richness and 0 for SIDI, SIEI, SHDI and SHEI), the range value is exclusively controlled by the maximum value. Range and maximum value from evenness and diversity indices are identical.

Range presents an identical evolution, since it is entirely defined by the maximum value, being for that reason a redundant and disposable parameter. These parameters progress along with the cell size, even when exhibiting significative distinct levels of values, so they cannot be considered a straightforward indicator of the optimal cell size.

The minimum value of richness does not vary among the eight cell sizes, and therefore does not provide any special information, except for cases when the minimum value equals the mode value, in the 1 km and 2 km cell sizes, which represent 65% and 46% of the total values, respectively (Table S3). These coincident factors can be good indicators for inappropriate cell sizes, especially if pondered with the multiplicity of mode. Effectively, at cell sizes of 1 km and 2 km, more than 80% of the total area is occupied by two single categories. Large maximum outliers (LM = median $\pm 3 \times$ IQR) are present between cell sizes of 1 km and 5 km, reflecting strong uneven distributions.

Mode of richness increases slowly, beyond the first two cell sizes, in general at a rate of one value per cell size, with the growth of cell area showing significative higher values in the two larger size cells (25 km and 30 km). Mode provides more information when combined with multiplicity of mode (Table S3). Multiplicity of mode decreases with the increasing of cell size, being less than 30% at a cell size of 5 km, where the mode value is 2, indicating that less than 30% of the cells are occupied by two distinct categories. Multiplicity of mode remains between 10% and 20% from a cell size of 10 km, suggesting a more diverse distribution of values.

IQR of richness presents the exact same values of mode until the cell size of 20 km, which reveals, in a general way, a very slight increase of dispersion within central values with the increase of cell size, mostly led by the third quartile, i.e., the highest central value, which also reflects an increase on the symmetry of richness distribution (Table S4). From the cell size of 20 km, IQR, increases slower than the mode, which does not demonstrate any particularity that could be of relevance for the determination of cell dimension.

In general, the maximum value (here represented by range) from both evenness indices (SIEI and SHEI) does not vary with the increase of cell size, not providing any information regarding cell size (Figure 3). Maximum evenness is attained from the 1 km cell size onwards, demonstrating complete independence of richness and of cell size. Diversity indices show a different evolution with the increase of cell size, reflecting the influence of richness. The maximum value from SIDI progresses slightly in the first cell sizes (1 km–5 km) and, in an almost imperceptible way, in the following cell sizes (10 km–30 km). The maximum value from SHDI evolves in a clearer way with the increase of cell size, revealing a strong influence of richness, progressing slightly faster between cell sizes of 1 km and 5 km than between cell sizes of 10 km to 30 km.

The minimum value (zero) of the evenness and diversity indices shows no variation among the eight cell sizes, providing no particular information, except that it also corresponds to the mode. As these indices are expressed in decimal values, zero becomes the most frequent value along the eight cell sizes for the four indices, reflecting the presence of a single category. Therefore, multiplicity of mode from the evenness and diversity indices (Table S3) gives straightforward information on the amplitude of no diversity or extreme dominance present in the area. Multiplicity of mode decreases with the increase in cell dimension, in a more accentuated way, between cell sizes of 1 km and 5 km (23%). At a cell size of 5 km, the zero value represents 23% of the total values, becoming less than 10% from a cell size of 10 km, reflecting a more diverse distribution of values (Table S4). Large maximum outliers are solely present in cell sizes of 1 km and within diversity indices (SIDI and SHDI), which is related to the strong dominance of the zero value.

The dispersion of central values varies with the cell size dimension, although it varies distinctly for the evenness and diversity indices, providing limited input concerning the optimal cell sizes (Figure 3). The IQR for diversity and evenness indices evolves very fast up to the cell size of 5 km, reflecting an increase of dispersion within central values, which is significantly conditioned by the decline of zero predominance, which, in turn, corresponds to the first quartile within the first two cell dimensions in all four indices. From the cell size of 5 km, the evenness indices exhibit a slight decrease along the following cell dimensions, which is mostly led by the increase in the first quartile values, i.e., the smallest central values (Table S4). The IQR from SIDI also decreases, but in an almost imperceptible way, showing some influence of richness. The richness influence is stronger in the IQR of SHDI, which evolves, with slight oscillations, from the cell size of 5 km, decreasing softly until a cell size of 15 km, and then increasing smoothly until the cell size of 30 km.

### 3.1.2. Quartile Coefficient of Dispersion and Coefficient of Variation

The quartile coefficient of dispersion and the coefficient of variation of richness present an overall smooth evolution along cell sizes, performed within a small range of values, especially when compared with the evenness and diversity indices, clearly showing smaller dispersion values within the first cell sizes (Figure 4). The coefficient of variation increases

between cell sizes of 1 km and 5 km, from the lowest value (0.44) to the highest value (0.53), slightly decreasing from there, maintaining dispersion values above the lowest value observed at a 1 km grid size. This clearly reflects the presence of large maximum outliers between cell sizes of 1 km and 5 km. The quartile coefficient of dispersion, being more robust, is almost invariable until a cell size of 5 km (0.33), slightly increasing in a cell size of 10 km (0.43), and slowly decreasing from that point until a cell size of 30 km (0.29).

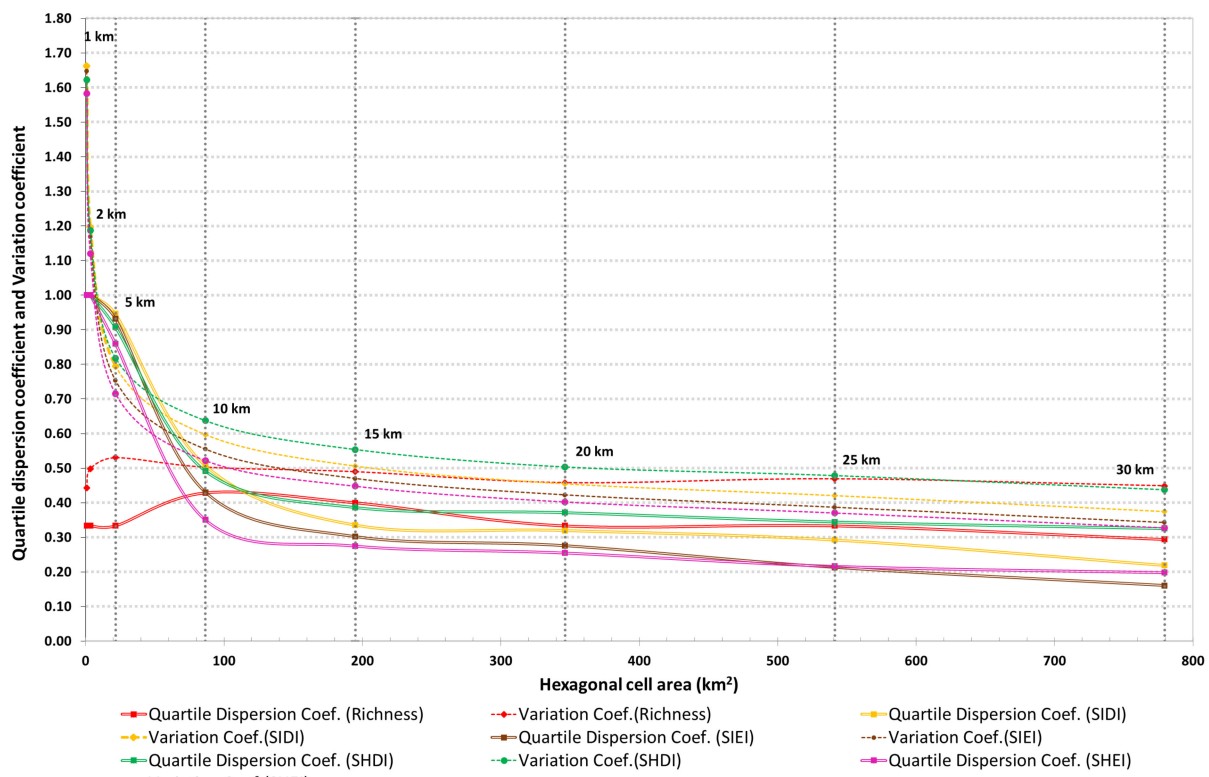

**Figure 4.** The quartile coefficient of dispersion (more robust) and coefficient of variation of richness, SIDI, SIEI, SHDI and SHEI calculated in a hexagonal analytical grid, along eight cell dimensions (1 km–30 km), intersected with lithology data from the geological map of mainland Portugal at a 1:1,000,000 scale.

The dispersion parameters from evenness and diversity indices sharply decline with the increase in cell size, particularly within smaller cell sizes, and may therefore be good indicators for optimal cell size(s) selection (Figure 4). An accentuated dispersion is present in the smaller grid sizes, decreasing in a very significative way until cell sizes of 5 km (coefficient of variation) and 10 km (quartile coefficient of dispersion), diminishing slowly from then with the increase in cell size. The coefficient of variation, being less robust to the presence of outliers, shows higher values—revealing standard deviation superior to the mean—than those from the quartile coefficient of dispersion, except within a cell size of 5 km, where it is slightly smaller. These extreme values are clearly related with dispersion of the higher values, induced by the predominance of zero value, which, in turn, is due to the significative presence of a single category throughout the map area within smaller cell grid sizes (1 km, 2 km). This can be confirmed by the observation of the histograms (Table S4). Additionally, in practically all cell sizes, there is a tendential sequence related with the dispersion values from the quartile dispersion coefficient and variation coefficient, exhibited by the evenness and diversity indices, from the highest to the lowest, respectively: SHDI (1.00–0.33; 1.62–0.44) > SIDI (1.00–0.21; 1.66–0.37) > SIEI (1.00–0.16; 1.66–0.34) > SHEI (1.00–0.19; 1.58–0.32).

### 3.1.3. Skewness Coefficient

The skewness coefficient from richness, SIDI, SIEI, SHDI and SHEI decreases with the increase in cell dimension, more sharply between cell sizes of 1 km and 5 km (Figure 5). The proximity to symmetry, or the distance from accentuated asymmetry, indicated by the skewness coefficient, could be a valid indicator of the optimal cell size(s).

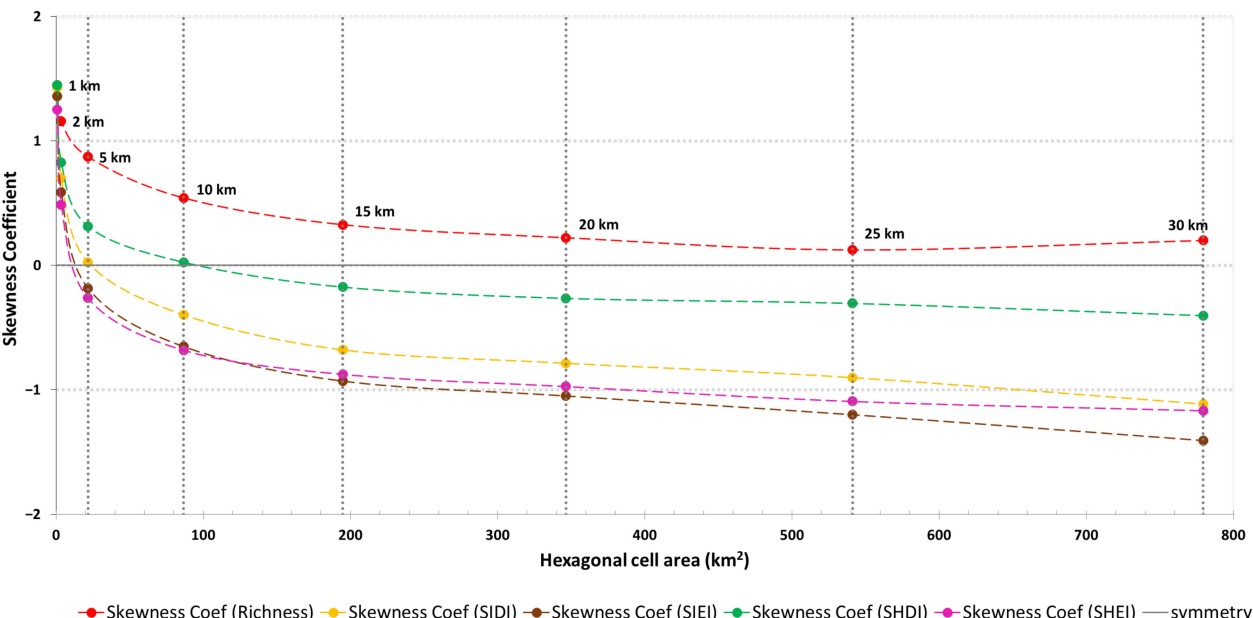

**Figure 5.** The skewness coefficient of richness, SIDI, SIEI, SHDI and SHEI calculated in a hexagonal analytical grid, along eight cell dimensions (1 km–30 km), intersected with lithology data from the geological map of mainland Portugal at a 1:1,000,000 scale.

Richness shows positive asymmetry through all the grid sizes analyzed, reflecting dispersion at higher values, being highly positively asymmetrical in cell sizes of 1 km and 2 km, and presenting, at a cell size of 5 km, almost half of the first cell value, decreasing softly from 10–15 km in size to values closer to symmetry, presenting the lowest values at a cell size of 25 km.

The evenness and diversity indices present highly asymmetrical values at a cell size of 1 km, which quickly decrease to values around 0 (symmetry) at a cell size of 5 km for SIDI, SIEI and SHEI, and at a cell size of 10 km for SHDI, before increasing slowly to more negative asymmetries from there. SHDI exhibits smooth progression almost parallel to the one observed for richness.

### 3.2. Geomorphological Indices–Statistical Parameters

3.2.1. Mode, Range, Min, Max and IQR

Even with distinct scale, polygon area distribution and number of total features (categories), when compared to the lithology database, the geomorphology data indicate strong similitudes related to the progress of basic statistics from richness, SIDI, SIEI, SHDI and SHEI along the eight cell dimensions examined (Figure 6).

Richness increases along with the cell size. The maximum value (or range) from richness progresses fast along with the cell sizes, stabilizing in the last cell dimensions (25 km and 30 km), peaking at a maximum value of 24 distinct categories. Although this platform of stable values within the last two cell sizes could constitute an indication of the optimal cell size, the clear vulnerability to cell size increase exhibited by this parameter means it cannot be considered a straightforward indicator of the optimal cell size.

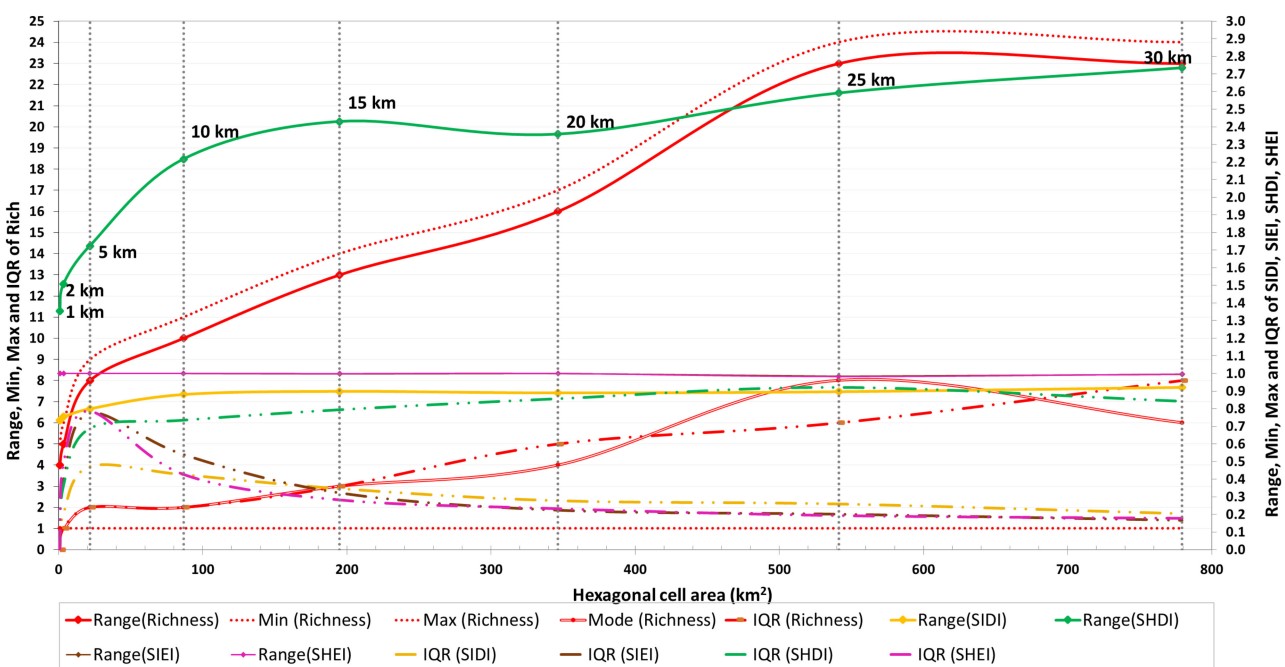

**Figure 6.** Mode (richness), range, min, max and interquartile amplitude (IQR) of richness, SIDI, SIEI, SHDI and SHEI, calculated in a hexagonal analytical grid, along eight cell dimensions (1 km–30 km), intersected with the geomorphological units' map of mainland Portugal at a 1:500,000 scale. SIDI, SIEI, SHDI and SHEI present much smaller values projected in the secondary axis of the graphic. Since the minimum value (min) is constant through all cell sizes for all indices (1 for richness and 0 for SIDI, SIEI, SHDI and SHEI), the range value is exclusively controlled by the maximum value. The evenness and diversity indices present identical range and maximum values.

The minimum value, mode and multiplicity of mode, especially when combined, can provide important information in this regard. The minimum value from richness equals the mode value, in the 1 km and 2 km cell sizes, representing 79% and 62% of the total values, respectively, indicating that these are unsuitable cell sizes (Table S5). Effectively, at cell sizes of 1 km and 2 km, more than 90% of the cells are occupied by two single categories. At a cell size of 5 km, mode, corresponding to two categories, represents 37% of the total values. Large maximum outliers are present between cell sizes of 1 km and 10 km, revealing strong positively asymmetric distributions. Mode of richness increases slowly along with the cell sizes, more sharply in the last cell dimensions, slightly decreasing in a cell size of 30 km. Multiplicity of mode decreases with the increase of cell size representing between 13% and 22% of the total values, from a cell size of 10 km, suggesting a more diverse distribution of values.

IQR from richness reflects an increase in dispersion within central values along with the growth of cell size, essentially led by the dominant increase in the third quartile, presenting extremely low values within cell sizes of 1 and 2 km.

Maximum values (represented by range) from evenness indices (SIEI and SHEI) do not vary with the increase in cell size, nor with the increase in richness. Maximum value from SIDI progresses slightly in the first cell sizes (1 km–10 km) and even more softly in the following cell sizes (15 km–30 km). Maximum value from SHDI evolves in a clearer way with the increase in cell size, revealing a strong influence of richness, progressing slightly faster between the cell sizes of 1 km and 10 km than between the cell sizes of 15 km to 30 km. Maximum values from diversity indices do not provide any relevant information for the optimal cell size(s) selection, per se, although the distinct rates of increase, particularly observed in SHDI, combined with other parameters could give some insight in this regard.

The minimum value (zero) of the evenness and diversity indices, also representing the mode value, shows no variation along the eight cell sizes. Multiplicity of mode from

evenness and diversity indices (Table S5) represents the amplitude of no diversity or extreme dominance occurring in the area. Multiplicity of mode decreases with the rise of cell dimension in a more accentuated way between cell sizes of 1 km and 5 km (31%). From a cell size of 15 km, those occurrences represent less than 7% of all values, reflecting more diverse distributions. Multiplicity of mode from diversity and evenness indices can provide pertinent information concerning the optimal cell size(s).

All diversity and evenness indices present large maximum outliers in a cell size of 1 km, and in cell size of 2 km for the diversity indices (SIDI and SHDI), which is also partially a consequence of the dominant presence of zero value.

As observed for lithology, the dispersion of central values varies along with the cell size, although, from a cell size of 5 km onwards, distinctly for SHDI and SIEI, SHEI and SIDI (Figure 6 and Table S6). The dispersion of central values increases very fast within the first cell sizes for the diversity and evenness indices, reflecting the progressive attenuation of zero influence. From a cell size of 5 km, the evenness indices and SIDI decrease slowly along the following cell dimensions, mostly led by the increase in first quartile values, i.e., the smallest central values. Richness influence is clear in the IQR of SHDI, which continues to increase smoothly until a cell size of 25 km. The dispersion of central values, particularly from the evenness indices, contributes significative input for the cell size(s) selection.

### 3.2.2. Quartile Coefficient of Dispersion and Coefficient of Variation

The quartile coefficient of dispersion and coefficient of variation from richness exhibit an evolution practically opposite to that presented by the evenness and diversity indices, and this evolution is also displayed within a smaller range of values (Figure 7). The quartile dispersion coefficient from richness strongly increases until a cell size of 5 km, achieving its maximum, then slightly decreasing within cell sizes of 10 km, increasing softly between cell sizes of 15 km and 30 km. Variation coefficient exhibits an almost parallel progression, increasing strongly between cell sizes of 1 km to 5 km (0.50), continuing to progress, more softly, until a cell size of 30 km. Both dispersion coefficients from richness present, from a cell size of 20 km, higher values than those observed in the evenness and diversity indices, being clearly more vulnerable to cell size increases.

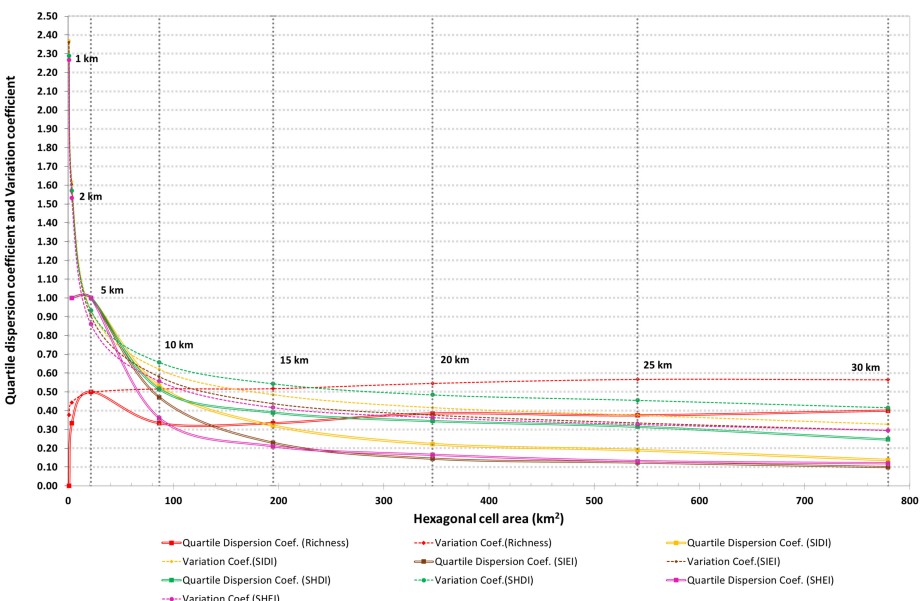

**Figure 7.** Quartile coefficient of dispersion and coefficient of variation of richness, SIDI, SIEI, SHDI and SHEI calculated in a hexagonal analytical grid, along eight cell dimensions (1 km–30 km), intersected with the geomorphological unit map of mainland Portugal at a 1:500,000 scale.

Dispersion parameters from SIDI, SIEI, SHDI and SHEI strongly lessen with the enlargement of cell size, in a very significative way between the cell sizes of 1 km and 5 km (coefficient of variation), and 10 km (quartile dispersion coefficient), progressing slowly within the higher cell sizes (Figure 7 and Table S6). The coefficient of variation presents much higher values than those from the quartile coefficient of dispersion (more robust), except within a cell size of 5 km. This is conditioned by the presence of zero value in the smaller cell grids, particularly in cell sizes of 1 km and 2 km, clearly indicating an inadequacy to provide proper spatial differentiation. As observed for lithology, the tendential sequence related to dispersion values (quartile dispersion coefficient and variation coefficient, respectively) can be identified from the evenness and diversity indices: SHDI (1.00–0.25; 2.29–0.42) > SIDI (1.00–0.14; 2.37–0.33) > SIEI (1.00–0.10; 2.36–0.30) > SHEI (1.00–0.12; 2.27–0.30).

### 3.2.3. Skewness Coefficient

The skewness coefficients from richness, SIDI, SIEI, SHDI and SHEI exhibit very analogous evolution, presenting highly positively asymmetrical in the first two cell sizes, decreasing with the increase in cell size, sharply until a cell size of 5 km, and more smoothly from then onwards, displaying very distinct values of symmetry (Figure 8). As previously noted, the closeness to symmetry, or the absence of extreme asymmetry indicated by the skewness coefficient, can help to support the optimal cell size(s) selection.

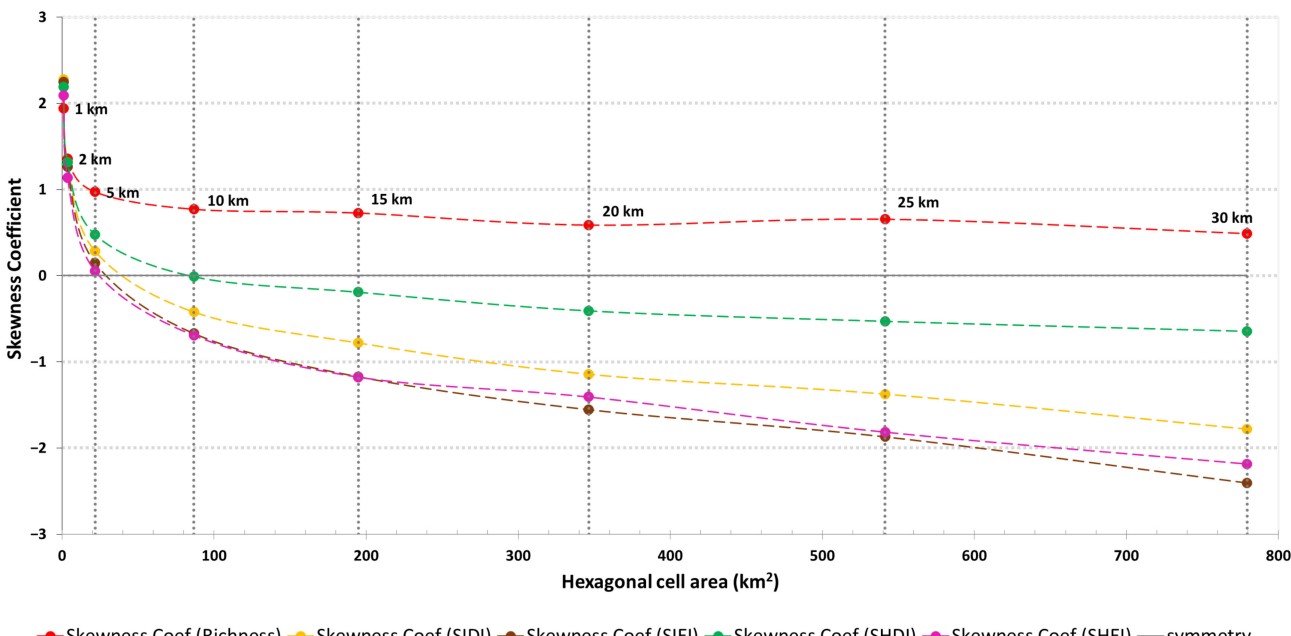

**Figure 8.** Skewness coefficient of richness, SIDI, SIEI, SHDI and SHEI calculated in a hexagonal analytical grid, along eight cell dimensions (1 km–30 km), intersected with the geomorphological unit map of mainland Portugal at a 1:500,000 scale.

Richness is the only index exhibiting positive asymmetry through all the grid sizes, reflecting dispersion at higher values, being highly positively asymmetrical in cell sizes of 1 and 2 km, present at a cell size of 5 km at approximately half of the first value, decreasing from then almost imperceptibly, never reaching symmetry.

The evenness and diversity indices present values close to symmetry, at a cell size of 5 km (SIDI, SIEI, SHEI) and at a cell size of 10 km (SHDI), from then decreasing progressively to negative asymmetries. The evenness indices present almost coincident evolution along the increasing cell sizes. As observed with lithology, SHDI exhibits smoother progression approximately parallel to richness.

### 3.3. Correlation Factors (Lithological and Geomorphological Indices)

The correlation factors of Spearman and Pearson show that, in general, correlation between the indices become less strong with the increase in cell size (Figure S2). In some cases, the first two cell sizes show extremely high correlation values (>0.9), then slightly decrease from a cell size of 5 km, occasionally oscillating in the higher cell sizes. These higher levels of correlation are probably enforced by the predominance of one to two single categories within the first two cell sizes.

The correlation factors also revealed stronger correlation values between the richness and diversity indices (SIDI (>0.7), SHDI (>0.8)) than with evenness indices (SIEI, SHEI). Furthermore, it is with SHDI that the strongest correlations between richness and diversity indices occur (in all cell sizes superior to 0.8 or to 0.9 in both datasets). Spearman's correlation factor, being more robust, shows higher values than Pearson's within the strongest correlated pairs of indices, and lower values in the weakest correlated pairs. The strongest correlated pairs are observed between SIDI–SHDI (>0.9), SIEI–SHEI (>0.9) and SIDI–SIEI (>0.8 or 0.9). The correlated pair in the diversity and evenness indices that shows lower correlation values is SHDI–SHEI, which is clearer in geomorphology dataset, showing moderated to very strong correlation values (>0.6 to >0.9).

The weakest correlations (0.3–0.5 and <0.3) occur between the richness and evenness indices (SIEI and SHEI). In these cases, it becomes very clear that the strongest values observed in the first cell sizes are inflected by the significative presence of one to two single categories. This is in accordance with the results presented in the previous section, and with the fact that evenness indices are independent of richness, contrarily to diversity indices that are, particularly SHDI, influenced by the number of distinct categories.

### 3.4. Lithological and Geomorphological Indices: Maps

Smaller cell sizes of 1 km and 2 km do not provide adequate spatial differentiation, because the corresponding indices' maps are dominated by very low to low classes, covering at least 60% (lithological) and 70% (geomorphological) of the total area (Figures 9 and 10). The reference maps (Figures S3 and S4) present very low to low values all over the area, with the higher values circumscribed to the borders between different categories.

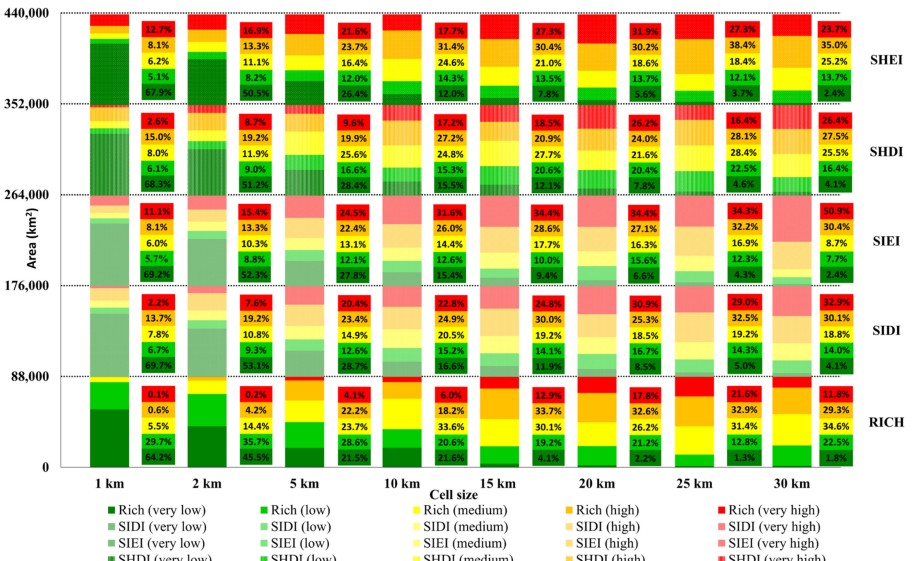

**Figure 9.** Synthesis of the lithological indices assessed for mainland Portugal, correspondent to 40 maps (eight cell sizes per richness, SIDI, SHDI, SIEI and SHEI). Five classes of diversity (very high, high, medium, low and very low) based on the Jenks natural breaks classification. Area (km$^2$ and %) is displayed per each of the five classes, the five indices and the cell size.

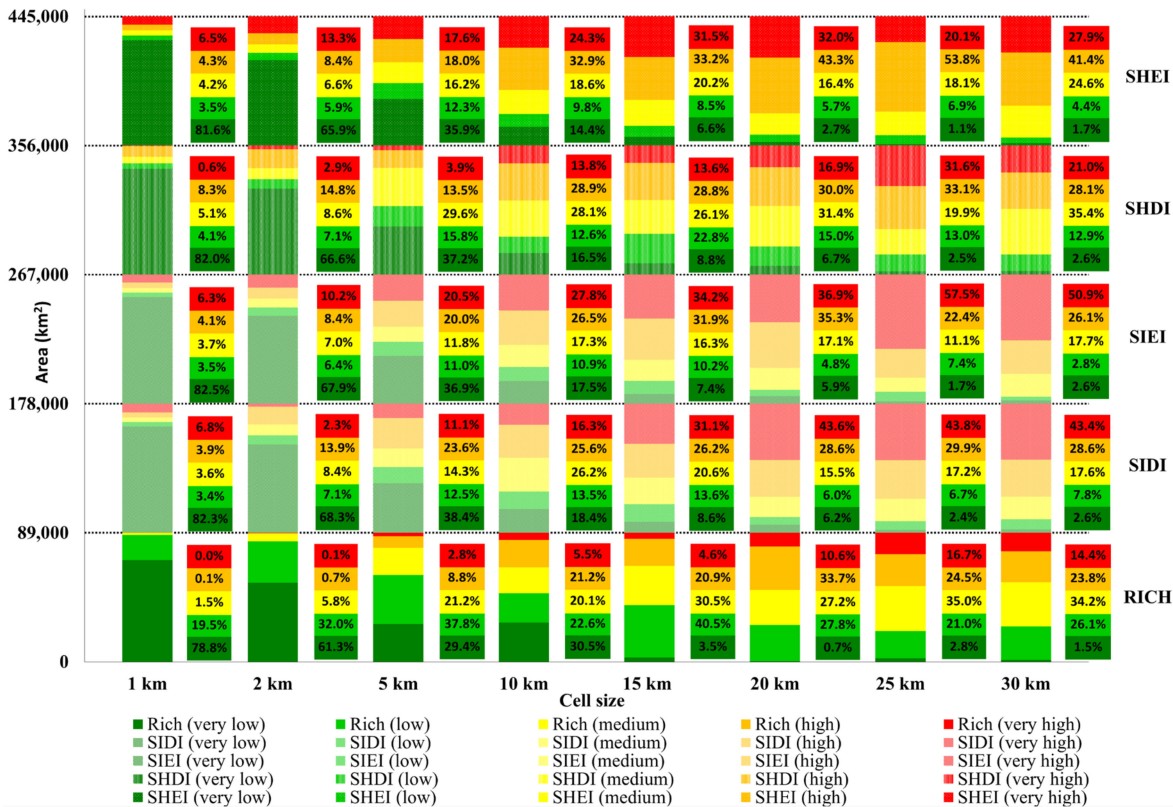

**Figure 10.** Synthesis of the geomorphological indices assessed for mainland Portugal, correspondent to 40 maps (eight cell sizes per richness, SIDI, SHDI, SIEI and SHEI). Five classes of diversity (very high, high, medium, low and very low) based on the Jenks natural breaks classification. Area (km$^2$ and %) is displayed per each of the five classes, the five indices and the cell size.

### 3.4.1. Lithological Indices: Maps

Regarding the lithological diversity dataset (Figure 9), the distribution of areas through the five classes become more even in the 5 km cell size, particularly for the evenness and diversity indices, although richness presents a quite even distribution of areas through the first four classes (very low to high), and a residual presence of the "very high" class (4%). The "medium" to "very high" diversity classes become dominant from cell sizes of 5 km for SHEI, SIEI and SIDI; in cell sizes of 10 km for SHDI; and in cell sizes of 15 km for richness. From that point on, with the increase in cell size, the distribution of area between the five classes becomes progressively more uneven, and the lower classes become residuals, leading to a smoother map, mainly characterized by three classes. From a cell size of 15 km onward, the "very low" class represents less than 12% for both diversity indices and less than 10% for both evenness indices, being quite residual (less than 5%) in the case of richness.

Considering the lithological indices' maps (Figure S3), and particularly the maps representing richness and SHDI, whose interpretation is more intuitive, some regional geological specificities can be identified within 5 km and 10 km cell sizes. Areas with higher diversity and richness are mainly associated with: (i) the very complex geological Galicia-Trás-os-Montes Zone (GTZ), characterized by the existence of the Bragança and Morais mafic and ultramafic polymetamorphic massifs, where large thrust systems make contact between the allochthonous complexes and associated para-autochthonous terrains, and between GTZ and Central-Iberian Zone (CZ); (ii) the shear zone that marks the transition between Ossa-Morena Zone and Central-Iberian Zone, which stretches from Oporto to Cordoba in Spain, passing by Tomar and Badajoz, (Blastomylonitic Belt), and also some smaller adjacent areas apparently related to the Lousã-Seia fault; (iii) the very complex and diverse Ossa-Morena Zone (OMZ), as a whole and particularly within the 10 km cell

size (SHDI), but in more detail (5 km) related to the Blastomylonitic Belt, the Estremoz–Barrancos Sector, in particular the Estremoz syncline, and the Montemor–Ficalho Sector, along the Ferreira–Ficalho thrust and the contact with the South-Portuguese Zone (SPZ), near the Pulo de Lobo formation; (iv) part of the Iberian Pyrite Belt in the South-Portuguese Zone (SPZ); (v) the Meso-Cenozoic basins, mainly the Lusitanian Basin areas related to the Nazaré fault and the Sintra and Arrábida Mountains. A slight contribution from the Ordovician Quartzitic Formation is reflected in the high values of diversity and richness within the Central Iberian Zone. Furthermore, the Messejana fault also contributes to the high values of diversity and richness defining an almost continuous line that crosses the SPZ and the OMZ.

Naturally, smaller cell sizes allow a more accurate identification of the regional specificities of structural indicators. Almost all these regional features are identifiable at a cell size of 2 km, although in a tenuous way. At a cell size of 10 km, some of these elements become smoother (the Nazaré, Messejana and Lousã-Seia faults, the Iberian Pyrite Belt and the Quartzitic Formation), which is even more evident while using SHDI instead of richness. SHDI, at a 10 km cell size, provides a map of high diversity in practically all of the OMZ, part of the SPZ and the GTZ, corresponding to more generalized results. With the increase in cell size above 10 km, the maps become even more generalized and aggregated, which results in the attenuation of regional differences, solely remaining the strongest elements, as the long shear zone that marks the transition between the OMZ and the CIZ.

3.4.2. Geomorphological Indices: Maps

Analyzing the geomorphological diversity dataset (Figure 10) in cell sizes of 5 km for SIEI and SHEI—and of 10 km for richness, SIDI and SHDI—the distribution of areas through the five classes become more even than within the first two cell sizes. From a 15 km cell size, the "very low" class represents less than 10% for the evenness and diversity indices, being quite residual (less than 5%) in the case of richness. "Medium" to "very high" classes become dominant from a 10 km cell size onwards, for the evenness and diversity indices, and from a 20 km cell size for richness. The 15 km cell size seems to define the point from which the distribution of areas between the five classes become progressively more uneven, reflecting a significative decrease in spatial differentiation with a more generalized map area, mainly characterized by three classes.

The geomorphological indices maps (Figure S4) show that the areas with higher diversity and richness are associated with: (i) the second hierarchical level unit of the NW Iberian Plateaus and Mountains in general, and specifically the third hierarchical level units of Peneda-Gerês mountains, the Peneda-Gerês Atlantic Front, the Mirandela tectonic basin and Strike-slip basins; the Push-up hills and the Alijó-Moimenta plateaus; (ii) the transition between the first level morphostructural units of the Cenozoic Basins and Slightly Deformed Meso-Cenozoic Basins with the Hesperian Massif, corresponding to part of the shear zone that marks the transition between the OMZ and the CIZ; (iii) the Lusitanian Basin (second level), specifically within the Estremadura limestone massif, part of the Caldas da Rainha limestone valleys and hills and of the Sicó-Alvaiázere hills, and also within the Nazaré-Peniche Coastal Plain; (iv) in the West and East Algarve Coastal Plains, in part of the SW Iberian Plateaus and Hills, within the Cercal-Caldeirão valleys and hills and Monchique mountain, and in the Algarve Basin within the Algarve Limestone Hills and Algarve Limestone valleys.

All of these local features are identifiable using the 5 km cell size grid, becoming clearer at a 10 km cell size, despite a certain loss of spatial differentiation due to the decrease in spatial resolution. At a 15 km cell size and above, most of these elements with highest diversity and richness show significative aggregation.

## 4. Discussion

### 4.1. Cell Size

All the performed analyses based on statistics and on map area distribution seem to indicate 5–10 km as the optimal cell size(s), even considering the differences between both datasets, namely distinct scale, distinct polygon area distribution and distinct number of total features (categories).

This conclusion is primarily based on the exclusion of the smallest cell sizes of 1 km and 2 km, which is considered inadequate due to the predominance of one to two single categories, with the higher values mostly confined to the borders between categories and associated with very high dispersion values and extremely asymmetric distributions. The use of these cell sizes would provide high resolution maps but generally underestimate diversity, with occasional overestimated spots on the intersections between three to four categories.

The cell sizes of 5 km and 10 km define the limit above which dispersion values from the evenness and diversity indices diminish in a significative way, and richness dispersion values evolve significatively more slowly, indicating more normal distributions. Additionally, at cell sizes of 5 km–10 km, richness distribution becomes less asymmetric, progressing from then slowly towards symmetry. More specifically, 5 km and 10 km cell sizes are associated with distributions closest to symmetry from the evenness and diversity indices, since SHDI shows, at a 10 km cell size, values close to symmetry; and SIDI, SIEI and SHEI, at a 5 km cell size, show the closest values to symmetry, evolving progressively from then to increasingly larger negative asymmetric values.

These results were confirmed and complemented with the analysis of the final maps of geological and geomorphological diversity based on the five indices and correspondent graphics that synthetize the area distribution (km$^2$ and %) per class. These graphics showed clearly, for both datasets, that map areas from first two cell sizes are largely dominated by "very low" to "low" diversity classes, confirming that these smaller cell sizes do not provide adequate spatial differentiation. Additionally, these graphics also show the limit above which the distribution of area between the five classes becomes progressively more uneven, and the lower classes become residuals, leading to a smoother map, mainly characterized by three classes. This limit can be variable, depending on the indices, although a cell size of 10 km seems to be the upper limit for richness, SIDI, SIEI and SHEI.

The maps produced revealed that various and distinct regional features associated with higher diversity and richness are identified using the 5 km and 10 km cell sizes (in fact, even with smaller cell sizes), above which maps become too generalized and aggregated with the increase in cell size, with loss of spatial differentiation which results in the attenuation of regional differences, and decrease in information, the strongest elements solely remaining, resulting in an overall overestimation of geodiversity. The geomorphology data used seems to be more adjustable to the 10 km cell size whereas, considering the lithology data, the appropriate cell size would be 5 km. Additionally, a smaller cell size provides more accurate results, provided it ensures sufficient spatial resolution to produce reliable values. The decision on the cell size should therefore be based on the data characteristics and the main purposes of the analysis. Furthermore, it can be of great relevance to proceed with distinct cell sizes and deeply analyze the effect of changing cell size on the geodiversity spatial pattern. Optimal cell size corresponds to the minimum size, once the dispersion values are significantly reduced or stabilized, and distributions from the evenness and diversity indices are closer to symmetry, which provides more accurate results and higher spatial differentiation, independently of range.

### 4.2. Statistic Parameters as Indicators

Range is frequently applied to define optimal cell size, while using the grid system [17,19,41,45,69], based on the principle that the most accurate differentiation of results is related to the maximum range between the highest and lowest diversity value. In this context, the proposal from Pereira et al. [17] defines the optimal cell size of the analysis

grid as the one that simultaneously provides the maximum range of values and the lower "minimum" value. Eiden et al. [41] selected the optimal cell size based on the maximum range of values which indicates simultaneously "the optimum spatial differentiation of the territory and the maximum range of diversity measures". In both methods, range is the determinant factor, although, in the proposal from Eiden et al. [41], the spatial differentiation of territory is equally considered. Although range could provide a direct measure of diversity, while using the grid system, it is also clear that its quality is strongly affected, and weakened, by its vulnerability to cell size increases.

In both the lithology and geomorphology datasets, the range taken from the five indices was entirely defined by the maximum value, and for that reason it is regarded as a redundant and disposable parameter. The maximum value of richness showed strong vulnerability to cell size increase, progressing in general along with the cell size. For that reason, it was not considered a straightforward indicator of the optimal cell size. For lithological indices, this parameter suggested two distinct platforms of stable values in the 10 km and 20 km cell sizes (for 11–12 categories) and in the 25 km and 30 km cell sizes (for 17 categories). For geomorphological indices, only one platform of stable values in the 25 km and 30 km cell sizes (for 24 distinct categories) was identified. If larger cell sizes were analyzed, the maximum value would most probably continue to grow and stabilize into platforms of higher values, although the minimum value would also increase, and the range would start to be distinct from maximum value.

Although more attenuated, a similar evolution was observed with SHDI, where the 10–15 km cell size marked a sharp change in the values increase in both datasets. These contrasts on the range/maximum value development could be used to identify the optimal cell size, combined with spatial differentiation provided by an adequate resolution. However, the differences observed in both datasets do not support a reliable conclusion, particularly when considering solely richness.

Furthermore, Bollati and Cavalli [45] used range to select the cell size, although this was conditioned to the average width of landforms polygons, tunning the limit above which cell size should be considered to ensure the maximization of diversity. This is a good complement, since it considers the fundamental characteristics of the base map. In both datasets analyzed in the present study, the average of polygon areas could not be used, as they both present considerable high coefficient of variation, especially the lithology dataset. The median from polygon area could be considered as the "confirmation" indicator for the selected size(s). For the lithology dataset, the median of the polygon area is 5.4 km$^2$, which is above cell area of 2 km (3.5 km$^2$) and below cell area of 5 km (21.7 km$^2$). The median of polygon area from geomorphology dataset is 71.8 km$^2$, below cell area of 10 km (86.6 km$^2$). These confirm the results identified by the dispersion and skewness parameters.

The skewness coefficient, quartile coefficient of dispersion and coefficient of variation provided more relevant, robust and straightforward information concerning the optimal cell size(s) selection. The dispersion parameters from the evenness and diversity indices sharply declined with the increase in cell size, particularly within smaller cell sizes, clearly and straightforwardly marking cell size(s) of 5 km (coefficient of variation) and 10 km (quartile coefficient of dispersion) as the size(s) from which the distributions become more normal.

The skewness coefficient provided a very straightforward indication for the optimal cell size(s), based either on the proximity to symmetry, or the distance from accentuated asymmetry, exhibited by the distributions from richness, SIDI, SIEI, SHDI and SHEI. Although these are the statistical parameters that gave the greatest indication for cell size selection, others also provided relevant data.

Even though the minimum parameter alone did not provide any information regarding the cell size, the coincidence with mode, solely observed in richness, was revealed to be a good indicator for exclusion of too small cell sizes, if considered with the multiplicity of mode. The multiplicity of mode from richness decreases with the enlargement of cell size, representing less than 30% (lithology) and 37% (geomorphology) at a 5 km cell size (mode value of 2), remaining between around 10% and 20% from a cell size of 10 km,

reflecting more diverse distributions. The multiplicity of mode of diversity and evenness indices directly measured the amplitude of no diversity or extreme dominance present in the area for each cell size, which decreases with increase in cell dimension, representing 23% (lithology) and 31% (geomorphology) of the total values at a cell size of 5 km, and less than 10% (lithology) and 13% (geomorphology) in a cell size of 10 km.

Dispersion on central values, IQR, from richness distribution, increased along the cell sizes, not providing any particularity that could be of relevance for the determination of optimal cell size(s), contrarily to IQR from evenness indices and, much less evidently, diversity indices, influenced by richness. These indices evolved very fast up to the 5 km cell size, reflecting an increasing of dispersion within central values, which is aligned with the decline of zero predominance, decreasing (except for SHDI) through the following cell dimensions, mostly led by the increase in the first quartile values, becoming more normal distributions.

### 4.3. Indices as Cell Size Indicators and Geodiversity Assessors

Richness is by far the most used index in geodiversity quantitative assessments, particularly while applying the grid system. It is also the less robust index due to its intrinsic vulnerability to cell size increase and is therefore unsuitable as a cell size indicator. In general, the evenness and diversity indices seem to be more qualified as cell size indicators when applying statistic parameters, particularly the evenness indices, which are not influenced by richness. Moreover, evenness indices clearly signaled the useful parameters of cell size indicators, as discussed in the previous section.

Richness provides clearer, understandable, and comparable results, while using maps to represent geodiversity. The areas with higher diversity and richness revealed in the maps produced corroborate the summaries displayed in Tables S1 and S2, related to the number of features per age, per geotectonic zone and correspondent area (for lithology), and per class within the distinct hierarchical levels of information (for geomorphology), respectively. Moreover, the results obtained for lithological and geomorphological diversity indices are generally in accordance with the results obtained by Peixoto [24], assessing the geodiversity of Portugal using a different grid size and format and a different geological base dataset.

Diversity indices maps, especially the ones resulting from SHDI, corroborate and emphasize the results from richness, which was expected, considering the high values of correlation exhibited by the two indices. Contrarily, the weakest correlations observed between richness and evenness indices suggest that both indices could provide complementary information, covering the compositional and structural components of diversity.

### 4.4. Other Methods and Applicability

Many authors applying the grid system adopt cell sizes without presenting the criteria underlying that decision. Many others adopt arbitrary cells sizes, based on published works that use input layers with similar scale and/or surface area. The most used method to select a cell size (either for the analysis grid and/or for the grid resolution), while using the grid system, is the simplest rule proposed Hengl [38], which is based on the scale of the input layers, e.g., [70–74]. Hengl [38] presented several rules to define a grid resolution based on cartographic, statistics and information theory concepts, some of which requiring advanced processing, and, for each type of rule, three standard grid resolutions (coarsest, finest and recommended) and correspondent formula were displayed. While applying the simplest rule based on the scale of the input layers to the layers used in the present study, the suggested coarsest cell size would be 2.5 km, if considering the smaller scale of 1,000,000 from the lithology dataset. This would be acceptable, although slightly too small to be use in the analysis grid, since that cell size is smaller than the cell size identified as optimal, according to the inherent characteristics of the input datasets and the immediate goal of the analysis, that is to assess diversity, as discussed in the previous sections.

The procedure proposed in this study considers simultaneously the properties of the input dataset and the goal of the analysis, resulting in the simplification of the input cartographic dataset. It is an empirical procedure, easy to apply, that tests diversity measurements within several cell dimensions, comparing and analyzing the results. Other authors used similar approaches, usually based solely on the range of richness, which was demonstrated in preceding sections to produce less robust results than the ones provided by dispersion (quartile coefficient of dispersion and coefficient of variation) and skewness coefficient measurements on diversity and, especially evenness indices.

Furthermore, the approach presented here is in accordance with and complements the medium-scale procedure proposed by Bartuś [30], which was strongly centralized in the analysis of the distributions from richness and SHDI. The empirical procedure here proposed, applied on a national scale, focused predominantly on dispersion analyses, and was based on the consubstantiate results from two distinct datasets (in scale, polygon area, number of polygons), and it is suitable for application to all type of scales and datasets. Nevertheless, it is important to highlight that the final selection is not a straightforward decision and might imply some adjustments to the function of the major goals of the analysis and the differences between datasets. In geodiversity assessment procedures, it is common to have different datasets from distinct origins, with distinct scales. Hence, it is important to stress how important it is to perform a previous analysis on the data in order to select more adequate cell sizes to further provide adequate estimations of geodiversity, as implied in the proposed procedure.

## 5. Conclusions

This work presents and tests, on Portugal's national scale, an empirical procedure to select optimal cell size(s), while using the grid system to calculate a geodiversity index, in lithology and geomorphology datasets. Richness, diversity and evenness indices were calculated, and several correspondent statistical parameters were measured. A special emphasis was given to dispersion parameters, considered, by far, more robust than the range.

The cell sizes 5 km and 10 km were identified optimal in this exercise, although the final decision should always consider the main purposes of the analysis. The optimal cell size corresponds to the minimum size once dispersion values are significantly reduced or stabilized, and distributions from the evenness and diversity indices are closer to symmetry. These attributes provide more accurate results, higher spatial differentiation and a reliable resolution.

The skewness coefficient, quartile coefficient of dispersion and coefficient of variation are the statistical parameters that provide more relevant, robust and straightforward information concerning the optimal cell size(s) selection.

The evenness and diversity indices showed more robust results than richness for cell size increase, and are therefore considered more qualified as cell size indicators when applying statistic parameters, particularly for the evenness indices. Richness provided clearer results in the final maps.

The selection of the cell size is determined for the accuracy and correctness of the final map results. Too small cell sizes produce good resolution maps but generally underestimate diversity, with occasional overestimated spots on the intersections between categories. Contrarily, too large cells generate overestimated geodiversity and coarse resolution maps. In geodiversity assessments using grid systems, a formal procedure for cell size selection does not exist. Many authors use arbitrary grid cells based on previous works or consider the scale of the input maps without complying with other inherent properties of datasets. Others consider less robust parameters, like range, and less robust indices, like richness, to assess the optimal cell size. The empirical procedure here presented is a methodological contribution applicable to all types of scales and datasets.

**Supplementary Materials:** The following supporting information can be downloaded at: https://www.mdpi.com/article/10.3390/resources12060065/s1, Figure S1: The lithostratigraphic map legend adopted and translated to English from Geological Map of mainland Portugal at a 1:1,000,000 scale produced by the National Laboratory of Energy and Geology [58,59,62]. The original can be found in https://geoportal.lneg.pt/media/p4wft3w5/cgp1m_2010.pdf (accessed on 12 May 2023). Table S1: Summary of number of features representing lithology (Singlefeature and Multifeature) per age and per geotectonic zone and correspondent area, used to assess the lithological diversity of mainland Portugal. Source: Geological Map of Portugal at a 1:1,000,000 scale [59].; Table S2: Summary of classes and number of features per class within the distinct hierarchical levels of information. Number of classes (NC) and number of features per class (NF), used to assess the geomorphological diversity of mainland Portugal. Total number of features: 686. Source: geomorphological units map of mainland Portugal at a 1: 500,000 scale (Figure 2b).; Table S3: Mode, multiplicity of mode and correspondent % of richness, SIDI, SIEI, SHDI and SHEI, used to assess the lithological diversity of mainland Portugal. Source: Geological Map of Portugal at a 1:1,000,000 scale [59].; Table S4: Histograms and boxplots of richness, SIDI, SIEI, SHDI and SHEI used to assess the lithological diversity of mainland Portugal. Outliers were identified by Median $\pm 1.25 \times$ IQR. Data were processed in Andad 7.12 (CVRM/IST, 2000). Source: Geological Map of Portugal at a 1:1,000,000 scale [59].; Table S5: Mode, Multiplicity of mode and correspondent % of richness, SIDI, SIEI, SHDI and SHEI used to assess the geomorphological diversity of mainland Portugal. Source: geomorphological units map of mainland Portugal at a 1:500,000 scale (Figure 2b).; Table S6: Histograms and boxplots of richness, SIDI, SIEI, SHDI and SHEI used to assess the geomorphological diversity of mainland Portugal. Outliers were identified by Median $\pm 1.25 \times$ IQR Data were processed in Andad 7.12 (CVRM/IST, 2000). Source: geomorphological units map of mainland Portugal at a 1:500,000 scale (Figure 2b).; Figure S2: Bivariate analysis with Pearson and Spearman correlation coefficients between richness, SIDI, SIEI, SHDI and SHEI, along eight distinct cell sizes, regarding geology and geomorphology datasets used to assess the lithological and geomorphological diversity of mainland Portugal.; Figure S3: Maps expressing lithological diversity, richness and evenness of mainland Portugal through five indices (richness, SIDI, SIEI, SHDI and SHEI) along eight cell sizes (1 km, 2 km, 5 km, 10 km, 15 km, 20 km, 25 km and 30 km). Each map represents five classes (very low, low, medium, high and very high) based on the Jenks natural breaks classification.; Figure S4: Maps expressing geomorphological diversity, richness and evenness of mainland Portugal through five indices (richness, SIDI, SIEI, SHDI and SHEI) along eight cell sizes (1 km, 2 km, 5 km, 10 km, 15 km, 20 km, 25 km and 30 km). Each map represents five classes (very low, low, medium, high and very high) based on the Jenks natural breaks classification.

**Author Contributions:** Conceptualization, C.L. and P.P.; datasets, C.L., D.I.P. and P.P.; methodology, C.L.; validation C.L.; writing—original draft preparation, C.L.; writing—review and editing, C.L., P.P. and Z.T.; supervision, P.P. and Z.T. All authors have read and agreed to the published version of the manuscript.

**Funding:** This research was funded by the European Social Fund (ESF) and the Portuguese funding institution FCT—Foundation for Science and Technology, grant number: UI/BD/150805/2020. This study had the support of FCT through the strategic projects UIDB/04683/2020 and UIDB/04292/2020 awarded to ICT and MARE and through project LA/P/0069/2020 granted to the Associate Laboratory ARNET.

**Acknowledgments:** The authors would like to acknowledge the National Laboratory of Energy and Geology (LNEG) for providing the geological cartography at scale 1:1,000,000.

**Conflicts of Interest:** The authors declare no conflict of interest.

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
