# Peer review of "Identifying Optimal Cell Size for Geodiversity Quantitative Assessment with Richness, Diversity and Evenness Indices"

_resources, doi:10.3390/resources12060065_

Round 1

Reviewer 1 Report

Dear editor and authors,

This is a certainly interesting paper, for it proposes a statistical method to decide the grid size for geodiversity calculation projects. The paper begins with a comprehensive review on the use of geodiversity calculation. Results, in spite of some statistical complexity, are clearly shown. English use is excellent. I recommend publication. 

Just one suggestion for a single sentence in page 3. Where it says "...the one providing the maximum range of values and the minimum "minimum" value" -which sounds odd-, I would suggest "the one providing the highest range of values and the lowest minimum value". 

Author Response

This reviewer proposed a single modification which was applied.

Reviewer 2 Report

Dear authors.

I have finished the review of the manuscript entitled: "Identification of the optimal cell size for the quantitative evaluation of geodiversity with indices of richness, diversity and uniformity". It presents a methodology to define an optimal cell size related to the scale of analysis, in turn linked to the importance of evaluating geodiversity as a basis for territorial management, planning of environmental and conservation strategies.

The manuscript is well written and organized, and meets standard scientific requirements. It has a long intro that I would shorten it to make it more eye-catching. The Methodology is clear and the Results are linked to it. However, I strongly suggest separating this last section using subheadings such as “Richness”, “Evenness” and “Diversity”, both when writing about “Lithological Indices” and “Geomorphological Indices”; this would make the results much clearer. The discussion and conclusions are appropriate. In addition to this, there is one point that should be fixed before considering that it can be accepted for publication. Figures 1 and 2 must be edited for the journal: the font is too small and the legend is illegible, and the text is not in English.

Overall, I recommend the publication of this manuscript after minor revision.

Minor editing of English language are required

Author Response

Since this reviewer proposed several modifications, the authors will reply point-by-point:

Reviewer: “The manuscript is well written and organized and meets standard scientific requirements. It has a long intro that I would shorten it to make it more eye-catching. “

Authors: Regarding this suggestion one sentence was deleted, but the general intro was kept, maintaining its consistency, considered fundamental to the work presented in the following sections.

Reviewer:“The Methodology is clear and the Results are linked to it. However, I strongly suggest separating this last section using subheadings such as “Richness”, “Evenness” and “Diversity”, both when writing about “Lithological Indices” and “Geomorphological Indices”; this would make the results much clearer.”

Authors: Regarding this suggestion, since the correspondent graphics, from Figure 3 to Figure 8, always represent the five indices, this suggestion would lead to a splitting of the graphics into multiple images (and there is already a significative number of images), otherwise the sections would be misplaced in regard to the reference images. Additionally in some cases, this would also lead to very small sections. Moreover, the authors consider pertinent the joint discussion with the five indices, although the more descriptive components are properly presented by distinct paragraphs. Nevertheless, and taking in consideration the suggestion by Reviewer 2, several subsections were created, based on the statistical parameters analyzed, both regarding “Lithological Indices” and “Geomorphological Indices”.

Reviewer:“The discussion and conclusions are appropriate. In addition to this, there is one point that should be fixed before considering that it can be accepted for publication. Figures 1 and 2 must be edited for the journal: the font is too small and the legend is illegible, and the text is not in English.”

Authors: The maps and correspondent legends were improved according to the suggestions from the reviewer 2. Figures 1 and 2 were replaced by new images. Since map legend from Figure 1 is very large, it is also presented as supplementary material, in larger size.

Further modifications by the authors

Additionally, in both Figures 4 and 7, the axis label was modified and the images were replaced. Some revisions of the text were made.
